# Profusion of G-quadruplexes on both subunits of metazoan ribosomes

**Santi Mestre-Fos[1,2], Petar I. Penev[1,3], John Colin Richards[1,2], William L. Dean[4], Robert D. Gray[4], Jonathan B. Chaires[4], Loren Dean Williams[1,2,3]***

**1** Center for the Origin of Life, Georgia Institute of Technology, Atlanta, Georgia, United States of America, **2** School of Chemistry and Biochemistry, Georgia Institute of Technology, Atlanta, Georgia, United States of America, **3** School of Biological Sciences, Georgia Institute of Technology, Atlanta, Georgia, United States of America, **4** James Graham Brown Cancer Center, University of Louisville, Louisville, Kentucky, United States of America

* loren.williams@chemistry.gatech.edu

**Data Availability Statement:** All relevant data are within the manuscript and its Supporting Information files.

## Abstract

Mammalian and bird ribosomes are nearly twice the mass of prokaryotic ribosomes in part because of their extraordinarily long rRNA tentacles. Human rRNA tentacles are not fully observable in current three-dimensional structures and their conformations remain to be fully resolved. In previous work we identified sequences that favor G-quadruplexes *in silico* and *in vitro* in rRNA tentacles of the human large ribosomal subunit. We demonstrated by experiment that these sequences form G-quadruplexes in vitro. Here, using a more recent motif definition, we report additional G-quadruplex sequences on surfaces of both subunits of the human ribosome. The revised sequence definition reveals expansive arrays of potential G-quadruplex sequences on LSU tentacles. In addition, we demonstrate by a variety of experimental methods that fragments of the small subunit rRNA form G-quadruplexes in vitro. Prior to this report rRNA sequences that form G-quadruplexes were confined to the large ribosomal subunit. Our combined results indicate that the surface of the assembled human ribosome contains numerous sequences capable of forming G-quadruplexes on both ribosomal subunits. The data suggest conversion between duplexes and G-quadruplexes in response to association with proteins, ions, or other RNAs. In some systems it seems likely that the integrated population of RNA G-quadruplexes may be dominated by rRNA, which is the most abundant cellular RNA.

## Introduction

rRNA expansion segments (ES's) decorate the surfaces of eukaryotic ribosomes. Ribosomal ES's of complex eukaryotes, especially in birds and mammals, contain long rRNA tentacles that appear to extend for 100's of Å from the ribosomal surface. We previously reported that rRNA tentacles of chordates can form G-quadruplexes [1]. rRNA tentacles of *Homo sapiens* contain multiple sequences that form G-quadruplexes of unusually high stability *in vitro*.

In our previous work, rRNAs were scrutinized for G-quadruplex sequences using the classic 3 x 4 motif of at least three contiguous guanines and a short spacer, repeated four times

**Funding:** Funded by LDW and JBC. Grant Numbers: LDW (80NSSC17K0295 and 80NSSC18K1139), JBC (GM077422). LDW (National Aeronautics and Space Administration). JBC (National Institute of Health and the James Graham Brown Foundation). https://www.nasa.gov/. https://www.nih.gov/. https://jgbf.org/. The funders had no role in study design, data collection and analysis, decision to publish, or preparation of the manuscript.

**Competing interests:** The authors have declared that no competing interests exist.

$[(G_{\geq3}N_{1-7})_{n\geq4}]$ [1]. A G-quadruplex is composed of four guanine columns surrounding a central cavity that sequesters monovalent cations ($K^+ > Na^+ > Li^+$) [2, 3]. The guanine columns are linked by Hoogsteen hydrogen bonds between co-planar guanines. The 3 x 4 sequence criteria identified four potential G-quadruplex forming regions in ES7 and ES27 of the rRNA of the large ribosomal subunit (LSU) of humans. rRNA of the human small ribosomal subunit (SSU) rRNA appeared to lack G-quadruplex sequences.

However, the RNA sequence space of G-quadruplexes has recently been re-evaluated: both shortened and bulged G-tracts are now seen to form stable G-quadruplexes [4–9]. Here, we extended our rRNA sequence search using a revised motif of four repeats of *two or more* adjacent guanines connected by a spacer (2 x 4, $(G_{\geq2}N_{1-7})_n$). Using the 2 x 4 criterion, we identify numerous additional G-quadruplex forming sequences on the human LSU rRNA, and for the first time, detect potential G-quadruplex forming sequences in SSU rRNAs (Fig 1). We experimentally confirm formation *in vitro* of G-quadruplexes by rRNA fragments derived from SSU rRNA as well as by native human SSU rRNA.

The revised sequence definition reveals expansive arrays of potential G-quadruplex sequences on LSU tentacles of expansion segments 7 and 27 (ES7 and ES27). The G-quadruplex forming regions of ES7 are increased from 10 to 20 G-tracts (tentacle *a*, regions 1 and 2) and from 4 to 23 G-tracts (tentacle *b*, regions 3 and 4) (Fig 1B). For ES27, the increases are from 6 to 25 G-tracts (tentacle *a*, regions 10 and 11) and from 0 to 32 G-tracts (tentacle *b*, regions 12, 13 and 14). rRNA sequences on the surfaces of the *H. sapiens* LSU and SSU that meet the 2 x 4 criteria are shown in Fig 1B and listed in Tables 1 (SSU) and 2 (LSU). G-quadruplex sequences in the SSU are located mainly near the termini of expansion segments (Fig 1B); es's are considerably smaller than ES's (lower case es indicates SSU and uppercase ES indicates LSU).

Here, we experimentally characterize SSU 2 x 4 G-quadruplex sequences of es3 and es12. Our data suggest that the 2 x 4 sequence of es3 forms a G-quadruplex *in vitro* while that of es12 forms a hairpin. The es12 sequence converts from hairpin to G-quadruplex in the presence of a G-quadruplex stabilizer or at elevated temperature. We also observe that 2 x 4 G-quadruplex sequences on the surface of the SSU are conserved over phylogeny of warm-blooded animals.

The combined results suggest that G-quadruplexes can be formed by multiple surface-exposed sequences on both the LSU and SSU of *H. sapiens*. The results are consistent with a model in which human ribosomal surfaces are structurally polymorphic, with complex liquid-liquid phase behavior, mediated in part by G-quadruplex formation. In metazoans, the integrated population of RNA G-quadruplexes (the RNA G-quadruplexome) may be dominated by rRNA, which is more abundant than other cellular RNAs [10, 11] and presents more expansive arrays of G-tracts.

## Results

### Computation

**Solvent accessible surface of the ribosome.** Here we presume that rRNA G-quadruplexes are more likely to form on the surface regions of the ribosome and not in the interior. The interior of the ribosome is engaged in intense inter- and intra-molecular interactions that constrain the structure and prevent conversion between different conformational states. To quantitate and visualize surface exposed rRNA, we calculated the Solvent-Accessible Surface Area (SASA) for all ribosomal nucleotides of the intact *H. sapiens* ribosome, and then mapped the data onto the two-dimensional structure of the rRNA (Fig 1A). The SASA varies between 0 Å$^2$ (fully buried) to 400Å$^2$ (fully exposed) per nucleotide. The color gradient shown in Fig 1A

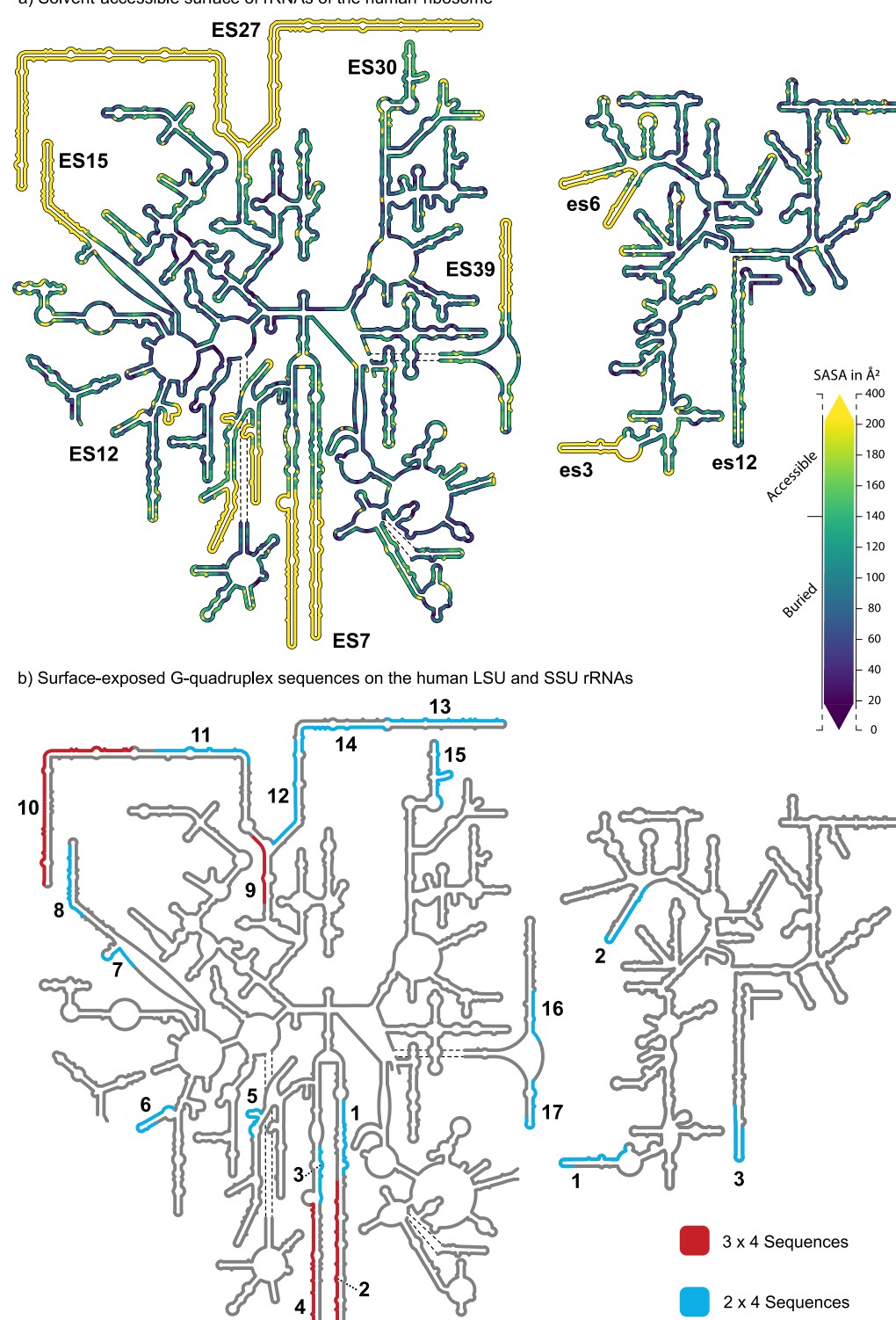

**Fig 1. Secondary structures of the LSU and SSU rRNAs of *Homo sapiens*.** (a) Nucleotide level solvent-Accessible Surface Area of the human LSU and SSU determined from intact ribosomes (PDB ID: 4UG0 [36]). (b) G-quadruplex forming sequences identified by the 3 x 4 motif are highlighted in red and those identified with the 2 x 4 motif are highlighted in blue. Only G-quadruplex forming sequences located on the ribosomal surface are shown. These images were generated with RiboVision [37].

**Table 1. Sequences and G-scores of 2 x 4 G-quadruplex regions located on the ribosomal surface of the *Homo sapiens* 18S rRNA (SSU).**

| Name | Region | SSU rRNA Sequence (5' to 3') | Highest G-score |
|---|---|---|---|
| GQes3 | es3 | **GG**CCCC**GG**CC**GGGGGG**C**GGG**C**GG**CC**GG**C**GG**CUUU**GG**[*] | 21 |
| GQes12 | es12 | **GGGG**UC**GG**CCCAC**GG**CCCU**GG**C**GG** | 20 |
| | es6 | **GG**AGC**GGG**C**GGG**C**GG**UCCGCCGCGA**GG** | 20 |

[*]GQes3 rRNA oligomer does not contain the last 9 nucleotides of the es3 sequence shown in Table 1

covers the range between 20Å$^2$ and 200Å$^2$ since the great majority of nucleotides fall in this range (S1 Fig). These ranges are consistent with previous definitions of exposed and buried nucleotides [12]. Nucleotides with SASA below 20Å$^2$ are purple, nucleotides with SASA above 200Å$^2$ are yellow. Nucleotides with SASA above 140Å$^2$ are considered as solvent accessible. Multiple contiguous nucleotides with SASA above 140Å$^2$ indicate rRNA that is on the ribosomal surface. Surface rRNA indicated by SASA was confirmed by inspection of the three-dimensional ribosomal structure. Nucleotides that are not resolved in the structure but are known to be on the surface are also yellow. These regions of the rRNA are of particular interest because conformational heterogeneity, possibly conversion between duplex and G-quadruplex forms, may be the ultimate source of the diffuse and smeared electron density during structure determination. Overall, the results confirm that rRNA expansion segments are in solvent-accessible regions of the ribosome.

**Identification of G-quadruplex sequences.** The 2 x 4 sequence definition significantly extends the repertoire of putative rRNA G-quadruplexes in the human rRNA. All 3 x 4 sequences are located on solvent-accessible ribosomal regions whereas several of the 2 x 4 sequences are buried in the interior of the ribosome. The interior 2 x 4 sequences would appear

**Table 2. Sequences and G-scores of 2 x 4 and 3 x 4 G-quadruplex regions located on the ribosomal surface of the *Homo sapiens* 28S rRNA (LSU).**

| Number | Region | LSU rRNA Sequence (5' to 3') | Highest G-score |
|---|---|---|---|
| 1 | ES7-a | **GG**C**GG**C**GGG**UCC**GG**CC**GG**GUGUC**GG**C**GG**CCC**GG**C**GG** | 20 |
| 2 | ES7-a | **GGGGG**C**GGG**CUCC**GG**C**GGG**U**GG**C**GGGGG**U**GGG**C**GGG**C**GGGG**CC**GGGGG**U**GGGG**UC**GG**C**GGGGG** | 60 |
| 3 | ES7-b | **GG**C**GGGGG**AA**GG**U**GG**CUC**GGGGGG** | 19 |
| 4[†] | ES7-b | **GGG**A**GGG**C**G**C**G**C**GGG**UC**GGG**GC**GG**C**GG**C**GG**C**GG**C**GG**C**GG**U**GG**C**GG**C**GG**C**GG**C**GG**C**GG**C**GG**C**GGG** | 38 |
| 5 | ES7-g | **GGG**CCC**GGGGG**A**GG**UUCUCUC**GGGG** | 19 |
| 6 | ES12 | **GG**CUCGCU**GG**C**G**U**GG**A**GG**CC**GGG**C**GG** | 20 |
| 7 | ES15 | **GG**AC**GGG**AGC**GG**C**GGGGG**C**GG** | 21 |
| 8 | ES15 | **GG**A**GGG**C**GG**C**GG**C**GG**C**GG**C**GG**C**GG**C**GGGGG**U**GG** | 21 |
| 9 | H63 | **GGG**CU**GGG**UC**GG**UC**GGG**CU**GGG** | 38 |
| 10[†] | ES27-a | **GGGGG**AGC**G**CC**G**C**G**U**GGGGG**C**GG**C**GG**C**GGGGGGG**AGAA**GGG**UC**GGGG**C**GG**CA**GGGG**CC**GG**C**GG**C**GG**CCC**G**CC**G**C**GGGG**CCCC**GG**C**GG**C**GGGGG**CAC**GG** | 40 |
| 11 | ES27-a | **GGGGGG**CCC**GGG**CACCC**GGGGGG**CC**GG**C**GG**C**GG**C**GG**CGACUCU**GG** | 37 |
| 12[*] | ES27-b | **GG**C**GGG**C**G**UC**G**C**GG**CC**G**CCCCC**GGGG**AGCCC**GG**C**GGG**C**G**CC**GG** | 20 |
| 13 | ES27-b | **GGGGG**C**GGGG**AGC**GG**UC**GGG**C**GG**C**GG**C**GG**UC**GG**C**GGG**C**GG**C**GGGG**C**GGGG**C**GG** | 21 |
| 14 | ES27-b | **GG**C**G**C**G**C**GG**C**GG**C**GG**C**GG**C**GG**CA**GG**C**GG**C**GG**A**GGG**CC**G**C**GGG**CC**GG** | 21 |
| 15 | ES30 | **GGGG**CCC**GGGG**C**GGGG**UCC**G**CC**GG**CCCU**G**C**GGG**CC**G**CC**GG** | 21 |
| 16 | ES39 | **GGG**ACC**GGGG**UCC**GG**U**G**C**GG** | 20 |
| 17 | ES39 | **GGG**AAAC**GGGG**C**G**C**GG**CC**GG**AGA**GG**C**GG** | 20 |

[*]This region contains a loop longer than 7 nucleotides

[†] These regions meet both 3 x 4 and 2 x 4 criteria.

to be locked in fixed non-G-quadruplex structures and were not investigated further. Canonical secondary structures of the human LSU and SSU rRNAs illustrating surface-exposed 2 x 4 and 3 x 4 sequences are shown in Fig 1B. Both surface-exposed and buried 2 x 4 and 3 x 4 sequences are shown in S2 Fig. Several of the 3 x 4 sequences in the human LSU have already been characterized [1]. Here, we concentrate on the 2 x 4 sequences of the human SSU rRNA, whose G-quadruplex forming capabilities have not been reported to our knowledge.

We identified three 2 x 4 sequences on the surface of the human SSU rRNA. These sequences are located in expansion segments es3, es6, and es12. The propensity of each of these 2 x 4 sequences to form G-quadruplexes was estimated with the program QGRS Mapper, which outputs G-scores [13]. The greatest SSU G-score (21) corresponds to the G-tracts of es3. Four 2 x 4 sequences are located on the buried interior of the ribosome. These interior 2 x 4 sequences are found in helices h33, h34, and in the junctions of helices h11 and h12 as well as helices h17 and h18 (S2 Fig).

G-tracts are 'polarized' in all LSU tentacles and in both es3 and es6 on the SSU. Polarized G-tracts are confined to one strand of the hairpin form of the tentacle with complementary C-rich sequences on the opposing strand (Fig 1B). In general, polarization is an indicator of proximal rather than dispersed G-tracts. Proximal G-tracts are more likely to form stable G-quadruplexes that dispersed G-tracts.

## Experiment

We produced RNA oligomers (GQes3 and GQes12, Fig 2A and S1 Table) that contain the 2 x 4 sequences found in tentacles of es3 and es12. We also investigated intact 18S rRNA (SSU rRNA) extracted from human cells. As negative controls, we produced mutants *mut*es3 and *mut*es12 with the same nucleotide composition as GQes3 and GQes12, respectively, but with disrupted G-tracts (G-scores: 0, S1 Table).

The G-tracts of es12 are distributed between both strands of the hairpin form of the tentacle (Fig 2A). We anticipate that in isolated oligonucleotides, polarized G-tracts are more likely to form G-quadruplexes than non-polarized G-tracts because for non-polarized G-tracts the hairpin form competes with G-quadruplex formation. Here, we experimentally investigate the 2 x 4 sequences of a polarized SSU tentacle (es3) and a non-polarized SSU tentacle (es12). The combined data, described below, suggest that GQes3 forms G-quadruplexes while GQes12 forms a hairpin at low temperature in the absence of G-quadruplex stabilizers. At elevated temperatures and in the presence of G-quadruplex stabilizers GQes12 converts from hairpin to G-quadruplex.

**Circular dichroism (CD) spectroscopy.** CD has been used extensively for inferring whether RNAs or DNAs form G-quadruplexes. RNA can form both parallel and antiparallel G-quadruplex topologies [14, 15]. The CD spectra of GQes3 and GQes12, with positive peaks at 260 nm and troughs at around 240 nm (Fig 2B), are consistent with RNA G-quadruplexes [16]. However, the spectra are not definitive because A-form DNA and G-quadruplexes present similar CD spectra [17]. The CD spectra of negative controls *mut*es3 and *mut*es12 are characteristic of A-C rich single-stranded RNA [18].

Thermal denaturation of GQes3 ($T_m > 95$ °C) or GQes12 ($T_{m,1} = 50$ °C, $T_{m,2} > 90$°C) was monitored by CD (Fig 2C) to reveal that both form exceptionally stable structures; neither is fully melted at the highest obtainable temperatures. Since neither completely unfolds, a detailed thermodynamic analysis is not possible.

When monitored at 265 nm the melting of GQes3 appears simple and is typical of G-quadruplex melting (Fig 2C and S3 Fig). However, the melting of GQes12 is complex, with a positive inflection near 50°C, followed by a separate denaturation transition with $T_m > 90$ °C. This

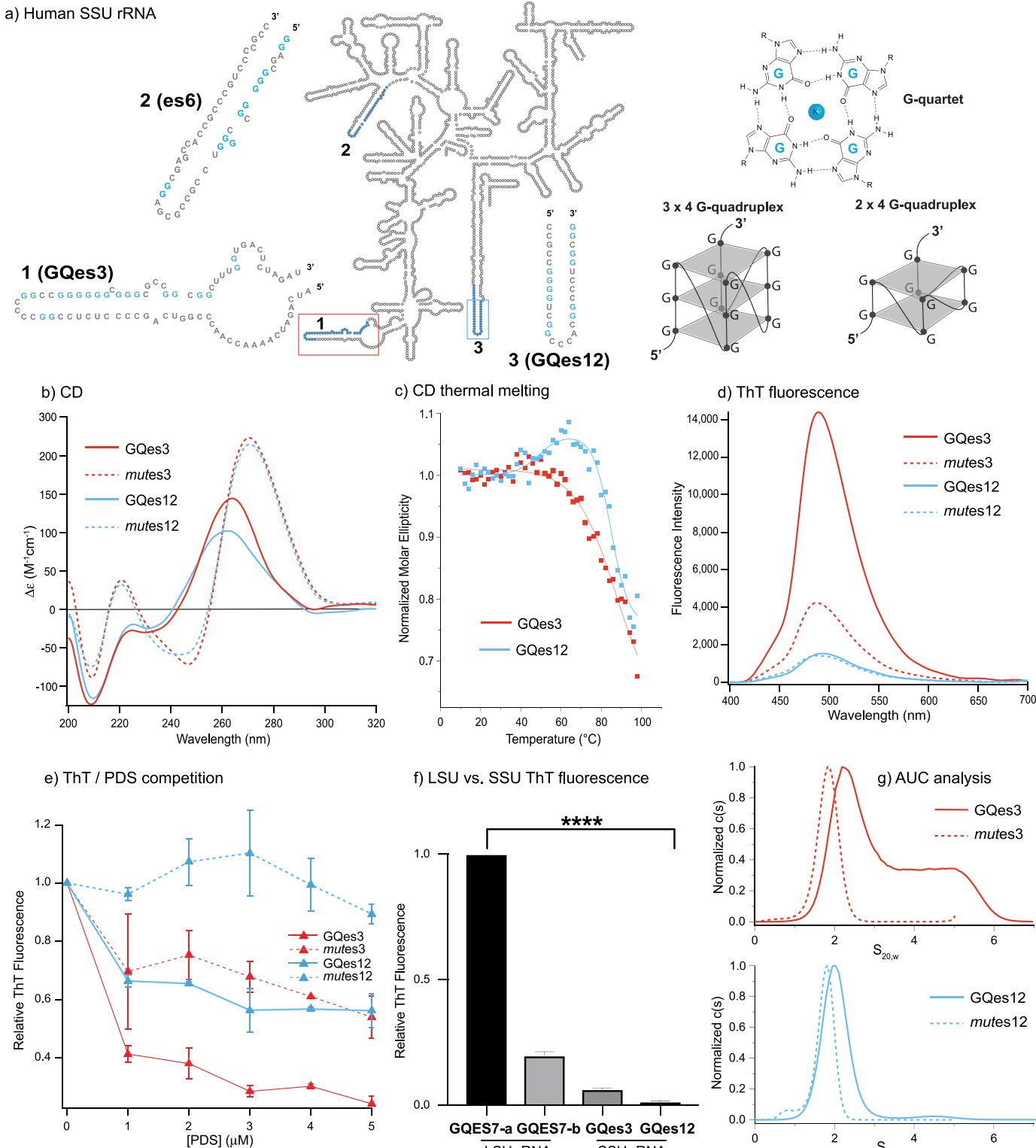

**Fig 2.** (a) **The** expanded secondary structure of the G-quadruplex regions on es3 or es12 of the human SSU rRNA. Schematic diagram of a single G-quartet, a 3 x 4 G-quadruplex (with tracts of at least three guanine residues), and a 2 x 4 G-quadruplex (with tracts of at least two guanine residues). (b) CD spectra of SSU G-quadruplex regions GQes3 and GQes12 and control RNAs *mut*es3 and *mut*es12. (c) Changes in CD amplitude at 260 nm as a function of temperature for GQes3 and GQes12. (d) ThT fluorescence in solutions of GQes3, GQes12, *mut*es3 or *mut*es12 annealed in the presence of potassium ions. (e) ThT/PDS competition assay. After annealing, the rRNA oligomers were incubated with increasing concentrations of PDS and then fluorescence was recorded at 490 nm. (f) ThT

fluorescence of LSU oligomers GQES7-a and GQES7-b and SSU oligomers GQes3 and GQes12. (g) Sedimentation velocities of GQes3, GQes12, *mutes3* or *mutes12*. Data are shown as c(s) plots that show the distribution of sedimentation coefficients obtained by analysis using the program SEDFIT [29]. The statistical significance relative to GQES7-a-ThT fluorescence is indicated by asterisks using an ordinary one-way ANOVA with Dunnett's post-hoc test. **** $P < 0.0001$.

biphasic behavior suggests that GQes12 may form one structure (perhaps a duplex hairpin, see below) at low temperature. Upon melting of the first structure, the strand appears to fold into a second structure (a G-quadruplex) that denatures with further increase in temperature. A probable hairpin structure for GQes12 is shown in S4 Fig (calculated using the mFold RNA server).

**Thioflavin T (ThT) fluorescence.** ThT interacts specifically with G-quadruplexes, and in doing so fluoresces at 490 nm [19]. Our results show that ThT in the presence of GQes3 gives a significantly stronger fluorescence signal than in the presence of the negative control *mutes3*, consistent with formation of G-quadruplexes by GQes3 (Fig 2D). However, GQes12-ThT fluorescence is not significantly higher than the negative control, consistent with formation of hairpin instead of G-quadruplex.

G-quadruplex stability is cation dependent ($K^+ > Na^+ > Li^+$). The cation dependence of ThT fluorescence is consistent with formation of G-quadruplexes by GQes3. ThT fluorescence is significantly more intense when annealed in $K^+$ than in $Li^+$ (S5 Fig). GQes12-ThT gives a stronger fluorescence in the presence of $Li^+$ than in $K^+$, consistent with our model in which GQes12 forms a hairpin and not a G-quadruplex.

**ThT—Pyridostatin (PDS) competition.** PDS is a G-quadruplex stabilizer that binds with greater affinity than ThT to G-quadruplexes and can displace ThT from G-quadruplexes [20]. The results here show that G-quadruplex-induced ThT fluorescence of GQes3 is attenuated by addition of PDS, consistent with displacement of ThT from G-quadruplexes by PDS. Although ThT fluorescence in the presence of either GQes3 or GQes12 decreases as PDS concentration is increased (Fig 2E), the effect on GQes3 is greater than the effect on GQes12. Under the conditions of our experiment, for GQes3 the maximum fluorescence intensity decrease is ~80% while the weaker signal of the *mutes3* control decreases by approximately 40%. For GQes12, the fluorescence intensity decreases by ~40% while the signal for *mutes12* control remains constant. The data are consistent with a model in which GQes3 forms G-quadruplexes in solution whereas GQes12 forms a hairpin in the absence of a G-quadruplex stabilizer. The G-quadruplex stabilizer shifts the GQes12 equilibrium to favor G-quadruplexes.

**LSU and SSU G-quadruplexes.** We compared the ThT fluorescence of 3 x 4 sequences from the LSU with 2 x 4 sequences GQes3 and GQes12 from the SSU, anticipating that the 3 x 4 LSU signals would be more intense than the 2 x 4 SSU signals. We previously characterized rRNA oligomers and polymers containing 3 x 4 sequences from the tentacles of *H. sapiens* LSU rRNA [1]. These sequences form highly stable G-quadruplexes *in vitro*. That work focused on 3 x 4 sequences within *tentacles a* and *b* of ES7. The LSU sequences are GQES7-a, sequence **1** in Table 2 and GQES7-b, the first four G-tracts of sequence **3** of Table 2. The G-scores of GQES7-a and GQES7-b are much greater than those of GQes3 and GQes12 (GQES7-a, 60; GQES7-b, 38; GQes3, 21; GQes12, 20).

The SSU sequences GQes3 and GQes12 give significantly weaker ThT fluorescence than the LSU sequences GQES7-a and GQES7-b. ThT fluorescence of GQES7-b is approximately 20% of that of GQES7-a, while GQes3-ThT fluorescence is around 6% and GQes12-ThT fluorescence is ~1.5% (Fig 2F). Overall, the experimental and computational results are self-consistent: the 3 x 4 G-quadruplexes in the human LSU rRNA appear to be considerably more stable and more extensive than the 2 x 4 G-quadruplexes of the SSU rRNA.

**Analytical ultracentrifugation (AUC).** The results of sedimentation velocity experiments (Fig 2G) are consistent with and support results described above. The control sequence *mut*es3, designed to be an unstructured single-strand, shows a single AUC species with an S value of 1.8 S. G-quadruplex sequence. GQes3 has a more complex c(s) distribution with major species of 2.5 S and 4.7 S (Fig 2G, **top panel**). The elevated baseline between these species suggests an interacting self-associating system, probably a mono-dimer equilibrium. The molecular weights of the two species observed for GQes3 indicates that they most likely correspond to monomer and dimer forms in equilibrium.

The difference in sedimentation coefficients between the GQes3 monomer and *mut*es3 indicates that monomeric GQes3 folds into a compact form, consistent with G-quadruplex formation. For comparison, the change in sedimentation of a DNA oligomer upon G-quadruplex formation is a similar to the change observed for GQes3. Specifically, the folding of a 24 nucleotide DNA oligomer into a G-quadruplex results in a change from 1.4S (unfolded) to 2.0S (G-quadruplex) [21]. Additional sedimentation velocity experiments (data not shown) showed that as the concentration of GQes3 is decreased, the fractional of 2.5S monomer increases, consistent with equilibrium between monomer and a dimer. The monomer-dimer concentration dependence of GQes3 mobility is confirmed by gel electrophoresis experiments (S6 Fig).

The AUC behavior of GQes12 (Fig 2G, **bottom panel**) differs from that of GQes3. The *mut*es12 control (unstructured) shows a single species with 1.75S, while under folding conditions GQes12 shows a single species of 2.03 S. S values are determined with 0.1S precision, so the difference is significant, but is smaller than expected for folding of GQes12 into a G-quadruplex. The data indicate folding of GQes12, presumably to a hairpin, which is less compact than a G-quadruplex.

**Association of SSU rRNA with BioTASQ.** G-quadruplex formation by the *H. sapiens* SSU rRNA was probed using BioTASQ, a biotin-linked small molecule that binds specifically to G-quadruplexes [22, 23]. BioTASQ associates with streptavidin beads and appears to provide an unambiguous assay for G-quadruplex formation; RNA that associates with BioTASQ is prevented by the beads from entering the gel during electrophoresis. As a negative control, we used *E. coli* SSU rRNA, which lacks GQes3, GQes12 or other potential surface-exposed G-quadruplex forming sequences. Our results indicate the *H. sapiens* SSU rRNA but not *E. coli* SSU rRNA binds to BioTASQ (Fig 3A) consistent with G-quadruplex formation by *H. sapiens* SSU rRNA. The simplest interpretation of this data, consistent with the other experiments presented here, is that *in vitro* GQes3 forms G-quadruplexes within *H. sapiens* rRNA.

We also investigated binding of GQes3 and GQes12 to BioTASQ. GQes3 binds well to BioTASQ while GQes12 binds poorly (Fig 3A). Treatment of cells with G-quadruplex stabilizers has been shown to significantly increase the efficiency of BioTASQ to pull down G-quadruplex forming RNAs [23]. Here, we observe that the addition of the G-quadruplex stabilizer PDS increases BioTASQ binding for all putative G-quadruplex forming rRNAs, including GQes12, but not for the negative controls. The results of these experiments suggest that GQes3 readily forms G-quadruplexes while GQes12 forms G-quadruplexes under stabilizing conditions. GQes3, GQes12 and possibly other SSU 2 x 4 sequences like the one found in es6 (Fig 2A) are probably responsible for the formation of G-quadruplexes within the intact *H. sapiens* SSU rRNA.

BioTASQ also binds human 28S rRNA but not *T. thermophilus* 23S rRNA (Fig 3A) which lacks surface-exposed G-quadruplex regions, indicating the formation of these secondary structures in the human LSU rRNA and corroborating our initial findings [1].

**Conservation of 2 x 3 sequences of the mammalian SSU rRNA.** We explored the phylogeny of G-quadruplex forming sequences in es3 throughout the Eukaryotic domain.

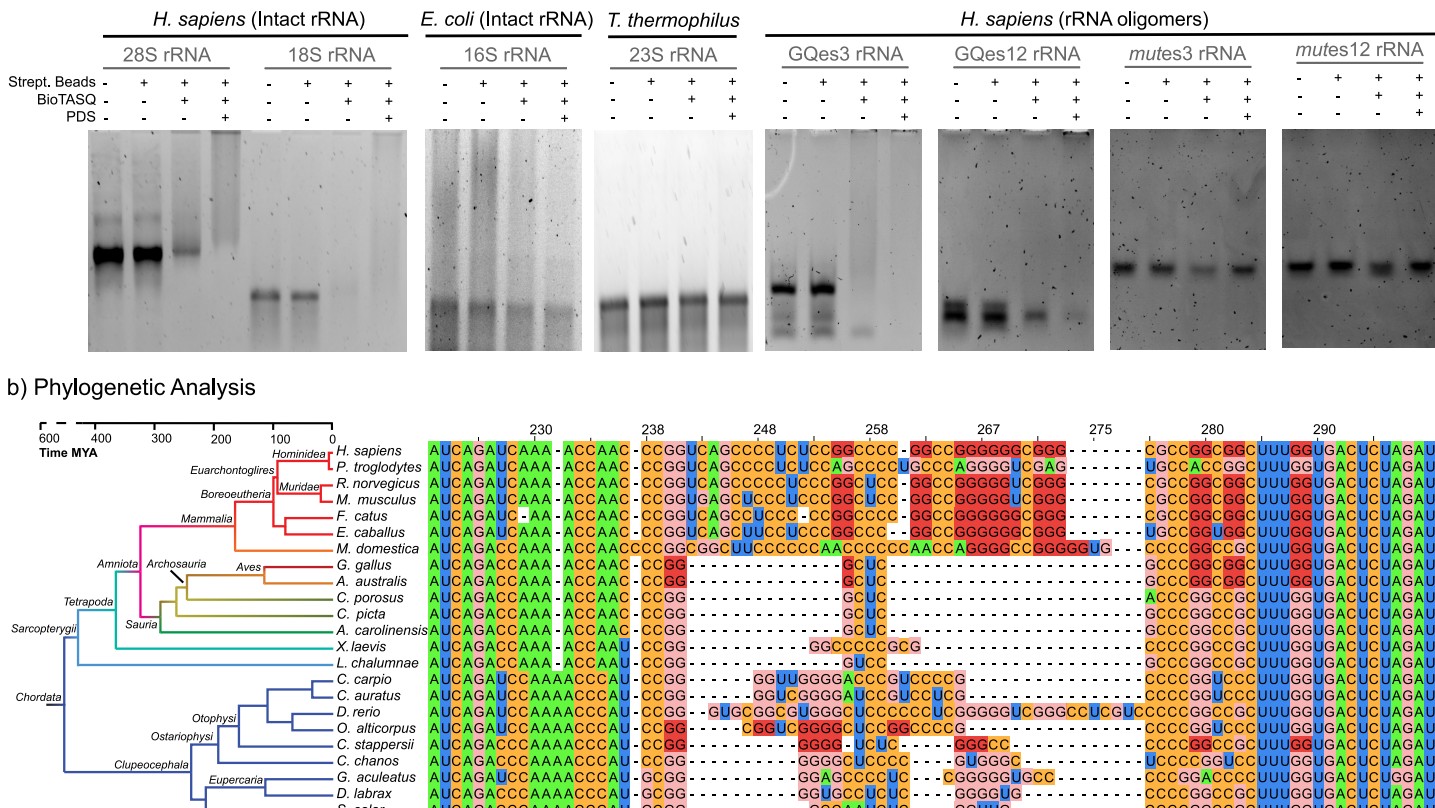

**Fig 3.** (a) BioTASQ binds to human LSU and SSU RNAs *in vitro* but not to *E. coli* or T. *thermophilus* rRNAs. BioTASQ binds G-quadruplex forming fragments of human SSU RNA but not to control RNA (b) Multiple Sequence Alignment of es3 for chordate species. G-tracts are common in chordates, specifically in mammals. Guanines from G-quadruplex forming sequences are highlighted with dark red, all other guanines are pink. All nucleotides are numbered in accordance with *H. sapiens* 18S rRNA. Human 28S (LSU), 18S (SSU) and *E. coli* 16S (SSU) rRNAs were extracted from cells. *T. thermophilus* 23S rRNA was synthesized *in vitro*. rRNAs were annealed in the presence of potassium and magnesium at pH 7.5, followed by incubation with BioTASQ and streptavidin-coated beads.

Multiple sequence alignments (MSAs) indicate that all mammalian species contain es3 sequences with the potential to form 2 x 3 G-quadruplexes (Fig 3B). G-tract polarization is conserved in mammalian es3 sequences. This pattern of conserved polarization was observed previously within expansion segments of the LSU. Using polarization as a marker for G-quadruplex formation, these results suggest that the last mammalian common ancestor had G-quadruplex forming sequences within es3.

**Parallel evolution of es3 within the ray-finned fishes.** Our initial SSU MSA included only a single representative of the ray-finned fishes (*Actinopterygii*)–*D. rerio*. The *D. rerio* sequence appeared anomalous in that es3 is long, comparable in length to es3 of mammals (Fig 3B). No 2 x 4 G-quadruplex forming sequences are observed in es3 of *D. rerio*. To investigate the evolution of es3 within the ray-finned fishes, we incorporated additional species to our SSU MSAs. The results show that es3 length is variable within ray-finned fish. Two of the ray-finned fish species in our alignment contain 2 x 4 G-quadruplex forming sequences. However, the 2 x 4 G-quadruplexes of ray-finned fish are not polarized.

The combined results suggest an uneven process of sequence evolution of es3 within the ray-finned fishes compared within mammals. The prospect of parallel evolution of G-quadruplex forming sequences between mammals and ray-finned fishes eliminates some possibilities for the underlying evolutionary pressures. The data limit the basis for G-quadruplex formation

to characteristics beyond those exclusive to mammalian lineages but rather to those that are common to all chordates.

## Discussion

Previously we identified sequences in the human LSU rRNA that form G-quadruplexes in vitro [1]. G-quadruplex sequences are a general feature of tentacles of chordate ribosomes. We identified G-quadruplex forming sequences in rRNA using the canonical 3 x 4 sequence motif. Sequences falling within this established motif were experimentally demonstrated to form G-quadruplexes in vitro. Recently however, others have shown that RNA sequences containing short tandem G-tracts that do not meet the 3 x 4 motif form G-quadruplexes [4–9]. A relaxed 2 x 4 criterion extends the repertoire of G-quadruplex forming regions, for example, to the UTRs of mRNAs encoding for polyamine synthesis proteins [4]. The development of next generation RNA sequencing strategies for mapping G-quadruplexes has significantly increased the ability to identify G-quadruplex sequences across the human transcriptome.

Here, applying the 2 x 4 criteria to human rRNA we identified seven potential G-quadruplex sequences in the SSU. Three of these are located on the ribosomal surface near the termini of the rRNA tentacles of es3, es6, and es12 (Fig 1). Our experimental results indicate that rRNA oligomers derived from these regions, as well as the native 18S rRNA, can form G-quadruplexes in vitro. This report represents the first evidence of G-quadruplex formation of any SSU rRNA.

The surfaces of both the LSU and the SSU of the human ribosome contain a sufficiently large number of G-quadruplex forming regions (Fig 4) that it seems possible that in some environments G-quadruplexes might dominate interactions of ribosomes with other cellular components. Extended arrays of G-tracts on solvent exposed regions of rRNA suggest roles in protein recruitment and polysome assembly. The association of the protein FUS [24, 25] with rRNA tentacles [1] is consistent with a model in which ribosomes participate in RNA mediated liquid-liquid phase separation [26].

Our phylogenetic analysis suggests that 2 x 4 sequences within the chordate SSU rRNAs have a complex evolutionary history. The data are consistent with a model in which 2 x 4

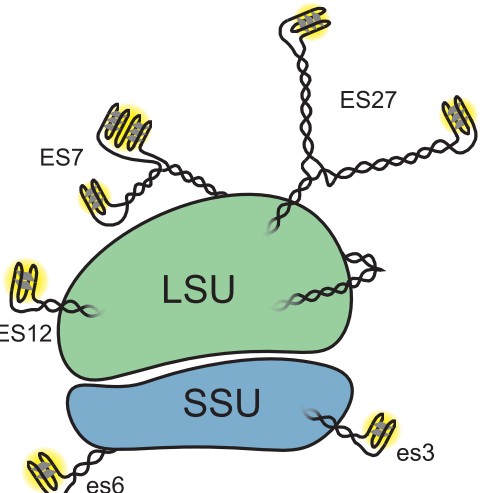

**Fig 4. Schematic representation of the *Homo sapiens* ribosome with G-quadruplexes.** ES lengths are not drawn to scale. This simplified schematic does not indicate the possibility of inter-ES and inter-ribosome G-quadruplexes or of the interconversion between G-quadruplexes and duplexes.

sequences evolved in parallel in distant chordate species. The phylogenetic conservation G-quadruplex sequences in warm-blooded animals and their surface localization suggest conserved function.

In yeast, ES27 has been shown to recruit specific proteins to the ribosome [27, 28]. In mammals, ES7 is extended by tentacles, reaching a size zenith in humans. Our results suggest LSU tentacles confer increased complexity compared with protists in part via ability to form G-quadruplexes. rRNA tentacles appear to be dynamic, switching between G-quadruplex and duplex conformations depending on environment or protein association.

rRNA makes up more than 80% of the cellular RNA, suggesting that most of the G-quadruplexes in cells at any given time could be contributed by rRNA and hence that the RNA G-quadruplexome could be ribosome-centered. Historically, study of RNA G-quadruplexes has been directed to poly-adenylated mRNAs and long non-coding RNAs. Most transcriptome-wide studies have explicitly excluded rRNAs from their analyses [9, 23]. The discovery of G-quadruplex regions on rRNA was made only recently [1]. The results presented in this study corroborate these initial findings and extend the G-quadruplex forming capability of rRNA to the *H. sapien*s SSU.

## Materials & methods

### RNAs

GQes3, GQes12, *mut*es3, and *mut*es12 were purchased from Integrated DNA Technologies. GQES7-a and GQES7-b were synthesized *in vitro* by transcription. *H. sapiens* 28S and 18S rRNAs were extracted from HEK293 cells. Briefly, HEK293 cells were grown to 60% confluency and total RNA was extracted with TRI reagent®. Intact rRNAs were extracted with a pipette from an agarose gel by running the rRNA into wells in the center of the gel. The rRNA was then precipitated with 5 M Ammonium Acetate-Acetic Acid (pH 7.5) with excess ethanol. *E. coli* 16S rRNA was extracted from DH5α *E. coli* strain using the same method. RNA sequences are listed in S1 Table.

### RNA annealing

Before any experiment, RNAs were annealed by heating at 95˚C for 5 min and then cooled to 25˚C at 1˚C/min and incubated for 10 min at 4˚C.

### CD spectroscopy

RNAs solutions were prepared at a final concentration of 10 μM (strand) and annealed as described above in the presence of 150 mM KCl and 10 mM Tris-HCl, pH 7.5. Spectra were acquired from 320 nm to 200 nm at a constant temperature of 20˚C on a Jasco J-815 spectro-polarimeter using 1-mm cuvettes, at a rate of 100 nm/min with 1-s response, a bandwidth of 5 nm, averaged over three measurements. The same buffer minus RNA was used as the baseline. Igor Pro software was used to smooth the data. The expression $\Delta\varepsilon = \theta/(32,980 \times c \times l)$, where c is the molar concentration and l is the cuvette path length was used to obtain the molar ellipticity from the observed ellipticity ($\theta$, mdeg).

### ThT fluorescence

RNAs solutions were prepared at a final concentration of 10 μM (strand) and annealed as described above in the presence of 150 mM KCl, 10 mM Tris-HCl, pH 7.5, and 2 μM ThT. For cation dependency experiments, either 50 mM KCl or LiCl was used. For LSU vs. SSU rRNA G-quadruplex-formation comparison (Fig 2f), RNAs were prepared at a final

strand concentration of 1 μM. After annealing, RNAs were loaded onto a Corning® 384 Well Flat Clear Bottom Microplate and fluorescence was recorded from 300 nm to 700 nm, exciting at 440 nm. Fluorescence data were acquired on a BioTek Synergy™ H4 Hybrid plate reader.

For ThT/PDS competition assays, PDS was added after RNA annealing at final concentrations of 1 μM, 2 μM, 3 μM, 4 μM, or 5 μM. Mixtures were allowed to sit at room temperature for 10 min before data acquisition.

## Analytical ultracentrifugation (AUC)

RNAs were prepared at a final $OD_{260}$ of 1.0 in the presence of 50 mM KCl and 10 mM Tris-HCl, pH 7.5. Sedimentation velocity measurements were carried out in a Beckman Coulter ProteomeLab XL-A analytical ultracentrifuge (Beckman Coulter Inc., Brea, CA) at 20.0 ˚C and at 40,000 rpm in standard 2 sector cells. Data (200 scans collected over a 10 hour centrifugation period) were analyzed using the program Sedfit [29] in the continuous c(s) mode or by a model assuming discrete, noninteracting species (www.analyticalultracentrifugation.com). Buffer density was determined on a Mettler/Paar Calculating Density Meter DMA 55A at 20.0 ˚C and buffer viscosity was measured on an Anton Paar Automated Microviscometer AMVn. For the calculation of frictional ratio, 0.55 mL/g was used for partial specific volume and 0.3 g/g was assumed for the amount of water bound.

## BioTASQ binding

RNAs solution were prepared at a final concentration of 15 nM (*H. sapiens* 28S, 18S rRNAs; *E. coli* 16S rRNA; *T. thermophilus* 23S rRNA) and 1 μM (GQes3, GQes12 or *mut*es3, *mut*es12) in the presence of 50 mM KCl, 1 mM $MgCl_2$ and 10 mM Tris-HCl, pH 7.5, and were annealed by heating at 75 ˚C for 1 min and cooling to room temperature at 1 ˚C/min. For (+) PDS samples, PDS was added to a final concentration of 5 μM after RNAs were annealed and allowed to sit at room temperature for 10 min. BioTASQ was added to the annealed RNA samples to a final concentration of 20 μM. Samples were allowed to mix by rocking at room temperature for 1 hr. Streptavidin-coated magnetic beads (GE Healthcare) were washed three times with 50 mM KCl and 10 mM Tris-HCl, pH 7.5. Then, 1.5 μg of the beads was added to the RNA-BioTASQ samples and allowed to mix overnight by rocking at room temperature. rRNAs were subsequently analyzed by native agarose electrophoresis (18S and 16S rRNAs) or 5% native-PAGE (GQes3, GQes12, *mut*es3 and *mut*es12).

## Phylogeny and multiple sequence alignments

The SEREB MSA [30] was used as a seed to align additional eukaryotic es3 sequences. The SSU rRNA sequences in the SEREB MSA were used to search [31] the NCBI databases [32] for SSU rRNA sequences. The SEREB database has sequences from 10 chordate species; sixteen additional chordate species were added to the es3 MSA (Fig 3). Sequences were incorporated into the SEREB-seeded MSA using MAFFT [33] and adjusted manually using Jalview [34]. Manual adjustments incorporated information from available secondary structures. In some cases, the positions of G-tracts in sequences with large gaps relative to H. sapiens are not fully determined, as they can be aligned equally well with flanking G-tracts in the MSA. Alignment visualization was done with Jalview [34]. The phylogenetic tree and the timeline of clade development were inferred from TimeTree [35].

## Supporting information

**S1 Fig. Solvent-accessible surface area for each nucleotide of a) human LSU rRNA and b) human SSU rRNA.**
(JPG)

**S2 Fig. Secondary structures of the human LSU and SSU rRNAs with 3 x 4 (red) and 2 x 4 (blue) G-quadruplex regions.**
(JPG)

**S3 Fig. Thermal denaturation of G-quadruplex forming fragments (A) GQes12, and (B) GQes3 monitored in circular dichroism.** The amplitude of the CD signal near 260 nm decreases as temperature is increased.
(JPG)

**S4 Fig. Possible hairpin structure of GQes12 calculated using the mFold RNA server.**
(JPG)

**S5 Fig. Relative ThT fluorescence spectra of the GQes3 and GQes12 rRNA oligomers annealed in the presence of potassium or lithium ions.**
(JPG)

**S6 Fig. Increasing concentrations of GQes3 result in a shift from the monomer to the dimer species.** Band intensities were quantified using ImageJ and the ratio of dimer to monomer was plotted. The increase in the ratio indicates that the equilibrium is shifted from monomer to dimer upon increase in the RNA concentration. RNA was resolved on a 6% Native PAGE.
(JPG)

**S1 Table. DNA and RNA sequences encoding RNAs used.**
(DOCX)

**S1 Raw Images.**
(PDF)

## Acknowledgments

The authors thank Sara Fakhretaha-Aval and Drs. Judy Wong and David Monchaud for helpful comments. BioTASQ was a generous gift of Dr. Monchaud. This work was supported by NASA: 80NSSC17K0295 and 80NSSC18K1139 (Center for the Origin of Life) to LDW and by NIH grant GM077422 and the James Graham Brown Foundation to JBC.

## Author Contributions

**Conceptualization:** Santi Mestre-Fos, Jonathan B. Chaires, Loren Dean Williams.

**Funding acquisition:** Jonathan B. Chaires, Loren Dean Williams.

**Investigation:** Santi Mestre-Fos, John Colin Richards, William L. Dean, Robert D. Gray.

**Resources:** Jonathan B. Chaires, Loren Dean Williams.

**Software:** Petar I. Penev.

**Supervision:** Santi Mestre-Fos, Loren Dean Williams.

**Validation:** Santi Mestre-Fos, John Colin Richards.

**Visualization:** Santi Mestre-Fos, Petar I. Penev, Jonathan B. Chaires, Loren Dean Williams.

**Writing – original draft:** Santi Mestre-Fos, Petar I. Penev, Jonathan B. Chaires, Loren Dean Williams.

**Writing – review & editing:** Santi Mestre-Fos, Loren Dean Williams.

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
