## [Decision Letter · Decision Letter 0]

22 Nov 2019

Profusion of G-Quadruplexes on Both Subunits of Metazoan Ribosomes

PONE-D-19-27856

Dear Dr. Williams,

We are pleased to inform you that your manuscript has been judged scientifically suitable for publication and will be formally accepted for publication once it complies with all outstanding technical requirements.

With kind regards,

Fenfei Leng, Ph. D.

Academic Editor

PLOS ONE

Additional Editor Comments (optional):

Reviewers' comments:

Reviewer's Responses to Questions

**Comments to the Author**

1. Is the manuscript technically sound, and do the data support the conclusions?

Reviewer #1: Yes

Reviewer #2: Yes

2. Has the statistical analysis been performed appropriately and rigorously? 

Reviewer #1: Yes

Reviewer #2: N/A

3. Have the authors made all data underlying the findings in their manuscript fully available?

Reviewer #1: Yes

Reviewer #2: Yes

4. Is the manuscript presented in an intelligible fashion and written in standard English?

Reviewer #1: Yes

Reviewer #2: Yes

5. Review Comments to the Author

Reviewer #1: This is an outstanding, thorough report investigating the quadruplex formed by RNA with sequences such as found in metazoan ribosomes. The experiments and results are described in a logical and systematic fashion. Further, the interpretations of the data are sound. This manuscript can be published as submitted.

Reviewer #2: This manuscript identifies G-quadruplex (G4) forming potential of certain rRNA tentacles from the small ribosomal unit of metazoans. This possibility has been explored using a number of methodologies, including computation of G-quadruplex forming potential, CD spectroscopy, melting potential, thioflavin fluorescence and thioflavin-pyridostatin competition, as well as analytical ultracentrifugation.

To some extent, using this multitude of methodologies to prove G4 is something of an overkill. Yes, I believe it--that under certain conditions these RNAs can form G4s. To me, it doesn't have to be shown in quite so many different ways. The more pertinent issue is what do these tentacles actually contribute to the ribosomal function? Any sign of their mediating clustering of ribosomes in vitro, for instance? Of the two tentacles considered, GQes3 has a better potential for forming G4s; the other (GQes12) appears to form a hairpin at low temperatures and has to be "pushed" by a templating substance to form a G4. All of this is good, and the data are acceptable.

My main concern is that this manuscript appears to to be little more than an addendum to these authors' much more substantial and provocative paper published earlier this year in JMB, which focused on the large ribosomal subunit. I don't see that this manuscript adds any incrementally newer ideas or provides really meaty new data relative to that earlier paper. So, while the data presented here are fine, does it all add up to a substantial enough contribution? It is up to the Editors of PLOS One to decide.

6. PLOS authors have the option to publish the peer review history of their article (what does this mean?). If published, this will include your full peer review and any attached files.

Reviewer #1: No

Reviewer #2: No

---

## [Editor Report · Acceptance letter]

6 Dec 2019

PONE-D-19-27856 

Profusion of G-Quadruplexes on Both Subunits of Metazoan Ribosomes 

Dear Dr. Williams:

I am pleased to inform you that your manuscript has been deemed suitable for publication in PLOS ONE. Congratulations! Your manuscript is now with our production department. 

With kind regards,

on behalf of

Dr. Fenfei Leng 

Academic Editor

PLOS ONE